# Can You Learn an Algorithm? Generalizing from Easy to Hard Problems with Recurrent Networks

**Avi Schwarzschild**
Department of Mathematics
University of Maryland
College Park, MD, USA
`avi1@umd.edu`

**Eitan Borgnia**
Department of Computer Science
University of Maryland
College Park, MD, USA

**Arjun Gupta**
Department of Robotics
University of Maryland
College Park, MD, USA

**Furong Huang**
Department of Computer Science
University of Maryland
College Park, MD, USA

**Uzi Vishkin**
Department of Electrical and Computer Engineering
University of Maryland
College Park, MD, USA

**Micah Goldblum**
Department of Computer Science
University of Maryland
College Park, MD, USA

**Tom Goldstein**
Department of Computer Science
University of Maryland
College Park, MD, USA

## Abstract

Deep neural networks are powerful machines for visual pattern recognition, but reasoning tasks that are easy for humans may still be difficult for neural models. Humans possess the ability to extrapolate reasoning strategies learned on simple problems to solve harder examples, often by thinking for longer. For example, a person who has learned to solve small mazes can easily extend the very same search techniques to solve much larger mazes by spending more time. In computers, this behavior is often achieved through the use of algorithms, which scale to arbitrarily hard problem instances at the cost of more computation. In contrast, the sequential computing budget of feed-forward neural networks is limited by their depth, and networks trained on simple problems have no way of extending their reasoning to accommodate harder problems. In this work, we show that recurrent networks trained to solve simple problems with few recurrent steps can indeed solve much more complex problems simply by performing additional recurrences during inference. We demonstrate this algorithmic behavior of recurrent networks on prefix sum computation, mazes, and chess. In all three domains, networks trained on simple problem instances are able to extend their reasoning abilities at test time simply by "thinking for longer."

## 1 Introduction

In computational theories of mind, an analytical problem is tackled by embedding it in "working memory" and then iteratively applying transformations to the representation until the problem is solved [Baddeley, 2012, Baddeley and Hitch, 1974]. Iterative processes underlie the human ability to solve sequential reasoning problems, such as complex question answering, proof writing, and even object classification [Liao and Poggio, 2016, Kar et al., 2019]. They also enable humans to extrapolate their knowledge to solve problems of potentially unbounded complexity, including harder problems than they have seen before, by thinking for longer.

35th Conference on Neural Information Processing Systems (NeurIPS 2021).

This work examines whether recurrent neural networks trained on easy problems can extrapolate their knowledge to solve hard problems. We find that recurrent networks can indeed generalize to harder problems simply by increasing their test time iteration budget (i.e., thinking for longer than they did at train time). Moreover, we find that the performance of recurrent models improves as they recur for more iterations, even without adding parameters or re-training in the new, more challenging problem domain. This ability is specific to recurrent networks, as standard feed-forward networks rely on layer-specific behaviors that cannot be repeated to extend their reasoning power.

The behavior we observe in recurrent networks falls outside the classical notions of generalization in which models are trained and tested on the same distribution. Because we train and test on problems of different sizes/difficulties, our training and test distributions are disjoint, and systems must extrapolate to solve problems from the test distribution. Outside the field of machine learning, computers achieve a functionally similar extrapolation ability through the use of algorithms, which encode the process required to solve a class of problems, and can therefore scale to problems of arbitrary size, albeit with longer runtime.

By training networks to solve problems iteratively, we hope to find models that encode a scalable *method* for solving problems rather than *memorizing* a mapping between input features and outputs. In short, the goal is to create recurrent architectures that are capable of *learning an algorithm*.

Our focus is on three reasoning problems that are classically solved using hand-crafted algorithms: computing prefix sums, solving mazes, and playing chess. Sequential reasoning tasks like these are ideal for our study because one can directly quantify the difficulty of a problem instance. In the case of mazes, for example, we can easily swap to a more challenging domain by increasing the size of the search space.

For each class of problems, recurrent networks are trained on a set of "easy" problems using a fixed number of iterations of the recurrent module on the forward pass. After training is complete, we assess whether our models exhibit logical extrapolation behaviors by testing them on "hard" problems, with varying numbers of additional iterations. Remarkably, models trained on easy examples exhibit little extrapolative behavior until their iteration budget is increased — generalizing to harder problems *requires* thinking deeper. Moreover, we find recurrent models tested with a sufficient number of extra iterations outperform the inflexible feed-forward models of comparable depth, often by a wide margin. Finally, we visualize the iterative behavior of the recurrent module to gain insights into the problem solving process they discover.

## 1.1  Related works

Our investigation into generalizing from easy to hard examples builds on several bodies of work. Logical extrapolation encompasses a special kind of distributional shift. A number of existing works on domain generalization instead explore shifts, such as re-stylization and image corruptions, which do not represent an increase in scale or computational complexity [Arjovsky et al., 2019, Shu et al., 2020]. Also, the basic neural architectures we use are not new and build upon prior studies of weight sharing and recurrence [Pinheiro and Collobert, 2014, Liang and Hu, 2015, Alom et al., 2018, Bai et al., 2018, 2019, Lan et al., 2020, Jaegle et al., 2021]. Networks with variable numbers of test time iterations/layers have also been studied, including variable depth networks [Graves, 2016, Huang et al., 2016, Kaya et al., 2019, Eyzaguirre and Soto, 2020].

Existing work on algorithm learning involves recurrent neural network (RNN) based approaches. For example, neural turing machines and neural GPUs can learn simple algorithms for tasks such as binary addition and multiplication [Graves et al., 2014, Kaiser and Sutskever, 2015]. Like most RNNs, the compute budget for these methods is inextricably tied to input length. Motivated by the fact that input sequence length is not necessarily correlated with the computational burden required to solve a problem, Graves [2016] develops a method for RNNs to adaptively select a compute time limit. This work considers only sequence inputs and shows the benefits of decoupling compute budget from input length. A differentiable extension of the technique can also be applied to visual question answering [Eyzaguirre and Soto, 2020].

The above works leverage neural networks with adaptive computation budgets to speed up and strengthen inference when learning on stationary distributions. In contrast, our work studies the logical extrapolation behaviors that recurrent networks possess when both computation budgets and problem difficulties are extended beyond the train-time regime. In the domain of constraint

satisfiability problems (CSPs), Selsam et al. [2018] show message passing neural networks trained on small CSPs can generalize to larger problems if more messages are passed at test-time – a similar type of extrapolation to our methods, but for a very specific problem formulation.

The particular problems we use to study this type of extrapolation include prefix sum computation and maze solving, two problems analyzed in the classical algorithms literature. For example, there are many ways to solve mazes, both classical (e.g. breadth first search) and learned (e.g. value iteration networks), but our goal is not to develop the best solver, rather to use mazes as a test bed for logical extrapolation [Tamar et al., 2016]. Our third case study is the game of chess, which has also been the focus of much artificial intelligence work [Romstad et al., Biswas and Regan, 2015, Silver et al., 2017, McIlroy-Young et al., 2020]. However, those efforts to play chess rely heavily on hand-crafted search algorithms, often paired with neural networks or opening books, and aim to play games from start to finish, both methods and goals that diverge from ours. As opposed to hard-coded algorithms, which scale by design, we are interested in studying whether learned processes can generalize from the data on which they are trained to even harder problems.

## 2  Dataset descriptions

We conduct experiments in three problem domains that are classically solved using hand-crafted algorithms. For each, we define datasets that have quantifiable notions of difficulty. This makes it possible to train models on easy/small examples and test them on harder/larger ones. We consider the task of computing prefix sums modulo two of binary bit strings, solving two-dimensional mazes, and finding the best move in chess puzzles.

**Prefix sums**   This problem is inspired by a similar dataset used by Graves [2016]. Each training sample is a binary string. The goal is to output a binary string of equal length, where each bit represents the cumulative sum of input bits modulo two. Our models accept input strings of any size, and we consider longer strings to be more difficult to process than shorter ones. Each dataset contains 10,000 uniform random binary strings without duplicates. We use datasets with input lengths of 32, 44, and 48 bits. See Figure 1 for an example input and the corresponding target output.

$$\begin{aligned} \text{Input:} \quad & [1, 0, 0, 1, 0, 1, 0, 1, 0, 1, 0, 1, 1, 0, 1, 1, 0, 1] \\ \text{Target:} \quad & [1, 1, 1, 0, 0, 1, 1, 0, 0, 1, 1, 0, 1, 1, 0, 1, 1, 0] \end{aligned}$$

Figure 1: Prefix sum input and target example.

**Mazes**   We consider mazes generated using a depth-first search algorithm [Hill, 2017]. We train on 50,000 small ($9 \times 9$) mazes, and we test on 10,000 larger ($13 \times 13$) mazes. Our models are convolutional, and receive a maze as a $N \times N$ three-channel image, where the maze walls are black, and the start and goal locations are red and green, respectively. The label for each maze is a binary two-dimensional mask containing the locations of positions along the shortest path solution. Our models output a two logits per pixel which are then thresholded to a binary mask of the whole input, and we consider a candidate solution correct only if it exactly predicts the labeled path. See Figure 10 for an example. More examples are available in Appendix A.1.2.

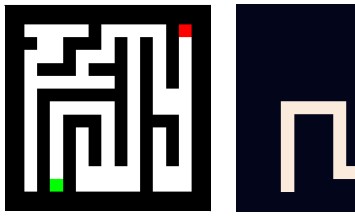
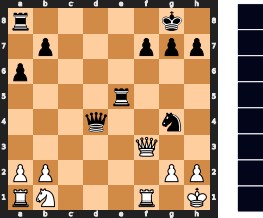
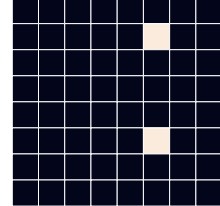

Figure 2: An example maze input and target. The target is a binary classification label for each pixel indicating on/off the optimal path.

Figure 3: An example of a chess puzzle input (left) and target (right). In the target, white represents one and black is zero.

**Chess puzzles** The third dataset we use is a corpus of chess puzzles. The data is furnished by Lichess, an online open-source chess server [Lichess, 2021]. From billions of games, Lichess compiles "puzzles" – mid-game boards for which a sequence of unique best moves is determinable (e.g. sequences of moves leading to a forced checkmate). From this database, we compiled labeled data where the inputs are $8 \times 8 \times 12$ arrays indicating the position of each piece on the board (one channel per piece type and color) and the outputs are $8 \times 8$ binary masks showing the origin and destination positions for the optimal move. See the example in Figure 3, where it is white to move and the solution to the puzzle is to move the queen from F3 to F7.

The puzzles each have a difficulty rating determined by an Elo-like system (a standard system for rating players using tournament play) [Elo, 1978]. When Lichess users, each with an Elo rating, attempt to solve the puzzle, the rating of the player and puzzle is updated as if they "played" against each other. After enough players encounter the puzzle, the rating of the puzzle reaches an equilibrium which gets recorded in the database. We use these ratings to distinguish between easy and hard; 600,000 puzzles of difficulty rating less than 1,385 are used for training, and testing is performed on 100,000 examples with ratings greater than 1,385.

Our models output a confidence score between zero and one for each square on the board. We take the two highest scores and compare their locations to the target to measure correctness – only exact matches are considered correct.

## 3 Model architectures & training

In all of the experiments in this work, we employ network architectures based on ResNets [He et al., 2016]. Our feed-forward networks are slight deviations from the most commonly used ResNets, in that the width does not change except at the first layer and after the last residual block, and we do not use batch normalization. This is done so that the recurrent models, whose internal recurrent module has the same input and output dimensions, can be as similar as possible to the feed-forward ResNets. In fact, the only difference during training between a feed-forward and recurrent model of the same effective depth is that the weights are shared between the residual blocks in the recurrent models. We refer to the recurrent portion of the network as the *recurrent module*. Also, our models are fully convolutional with no fully connected heads. For solving prefix sums, which involves one-dimensional strings, we further deviate from classical ResNets by using one-dimensional convolutions. For complete architectural details, see Appendix A.2.

We measure recurrent models in terms of *iterations* and *effective depth*. An iteration is a repetition of the recurrent residual block, which contains four layers in all of our models. Therefore, the effective depth is equal to four times the number of iterations, plus non-recurrent encoder and head layers that sandwich the recurrent module. For example, the models used for computing prefix sums have one convolutional encoder layer, followed by the recurrent block and then by a three layer convolutional head. In this case, a 10-iteration model has effective depth $1 + 10 \times 4 + 3 = 44$ layers.

For each training sample we consider, the label is an array of binary classification variables, one per input pixel. The training loss is simply the mean cross-entropy loss across output values. In general, hyperparameters were determined with the goal of finding convergent models. The specific batch sizes, learning rate and decay schedules, and other hyperparameter values are all available in Appendix A.3. [1]

## 4 Recurrent networks can generalize from easy to hard problems

We explore the ability of recurrent neural networks to generalize to more difficult problems simply by thinking deeper. To this end, we train models of varying effective depth on easy training examples and test them on harder problems. We find that recurrent models are even better at generalizing from easy to hard than their feed-forward counterparts. While there is only one way to test the feed-forward models, we take a closer look at what happens when the recurrent models are allowed to think deeper about the harder problems. Formally, we use more iterations of the recurrent module within the recurrent models when performing inference on test data. Across all three problem types, we find that

---

[1] Code to reproduce our experiments along with information about downloading the data we use is available at https://github.com/aks2203/easy-to-hard

the confidence of the model is a good surrogate for correctness. Therefore, when evaluating recurrent models, we use the output from the iteration to which the network assigns the highest confidence.

## 4.1 Prefix sums

The first task on which we demonstrate the ability of recurrent neural networks to learn an algorithm is one from the classical algorithms literature, namely computing prefix sums. Specifically, we study the problem of computing the prefix sums modulo two of binary input strings.

When computing prefix sums, we employ models with effective depths from 40 to 68 layers. We train models on easy data consisting of 32-bit input strings and test on harder 40-bit and 44-bit strings. In Table 1, it is clear that even when holding the depth constant at test time, recurrent models generalize from easy to hard better than feed-forward networks.

Table 1: **Extrapolating to longer input strings.** Shown here are the average accuracies of models trained on 32-bit inputs and tested on 40-bit inputs. The effective depths listed below correspond to 9, 10, and 11 iterations in recurrent models. We report average accuracy $\pm$ one standard error.

| | Effective Depth (Layers) | | |
| | 40 | 44 | 48 |
|---|---|---|---|
| Recurrent | $24.96 \pm 2.96$ | $31.02 \pm 2.56$ | $35.22 \pm 3.34$ |
| Feed-forward | $22.17 \pm 0.85$ | $24.78 \pm 1.65$ | $22.79 \pm 1.32$ |

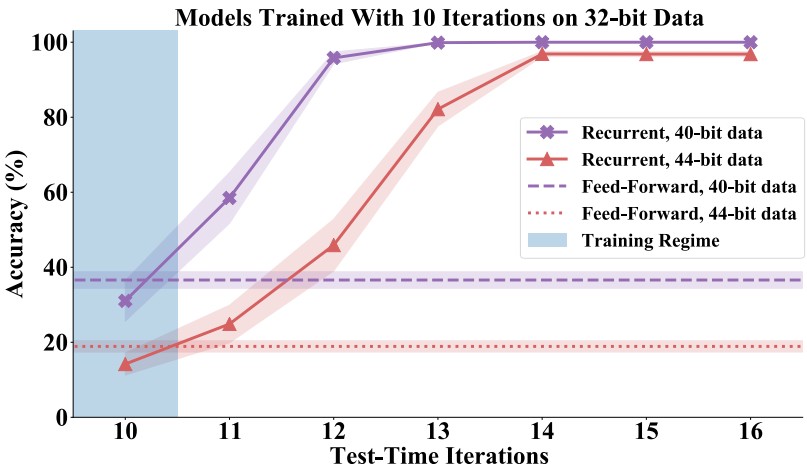

Figure 4: Generalizing from easy to hard prefix sums. The ability of networks to compute prefix sums on two test sets with longer input strings than were used for training (accuracy on 40-bit inputs in purple and on 44-bit inputs in red). We compare recurrent models to the best feed-forward models of comparable effective depth. The markers are at average values from several trials and the shaded regions indicate $\pm$ one standard error.

When the thought budget, or number of iterations, is increased, we see that recurrent models can get upwards of 90% of the harder testing examples correct. In Figure 4, we observe this large boost in the recurrent models' performance and a vast difference in the accuracy of recurrent models (with added iterations at test time) and feed-forward networks. Note that the dotted lines represent the average accuracy of the deepest feed-forward networks considered. That depth is 68 layers, or the effective depth of recurrent models with 16 iterations, and we use this baseline in the plot specifically because in the range of depths corresponding to the numbers of iterations shown, these feed-forward models achieve the highest accuracy. In other words, recurrent models trained with relatively few iterations generalize well to harder data while similar and even much larger feed-forward networks fail to generalize in the same scenario.

The generalization we see in Figure 4 indicates that these recurrent models learn processes that can be extended to harder problems by running for more iterations. In particular, the recurrence is both

the machinery that allows for varying the depth at test time, as well as a force at training to push the model to find parameters that make progress toward a solution with each reuse.

When these results are viewed through the lens of algorithm design, one might wonder how the *receptive field*, or the number of entries in the input that determine a single entry in the output, affects these models. The feed-forward models, whose accuracies are shown in Figure 4, have the same receptive field as the recurrent models when tested with 16 iterations. This makes it clear that the increase in accuracy of recurrent models does not simply occur because the receptive field grows with added iterations, rather it occurs because they have learned a process that can extrapolate beyond the training distribution. Further discussion on receptive field is presented in Appendix B.

### 4.1.1 Iterative outputs

One way to dissect the learned process and compare it to known algorithms is to plot the confidence of the model at each iteration. In Figure 5, we show a representative example of a network's confidence that each bit in the output is a one. Two striking observations can be made from this figure. The first is that the model is progressing to the solution with each iteration. The second observation is that it is resolving the prefix sum in the earlier bits first and moving down the string, settling on the final bits only in the last iteration. This is remarkably similar to a naive algorithm one might implement for this task that marches from the first index until the end of the string computing prefix sums in order.

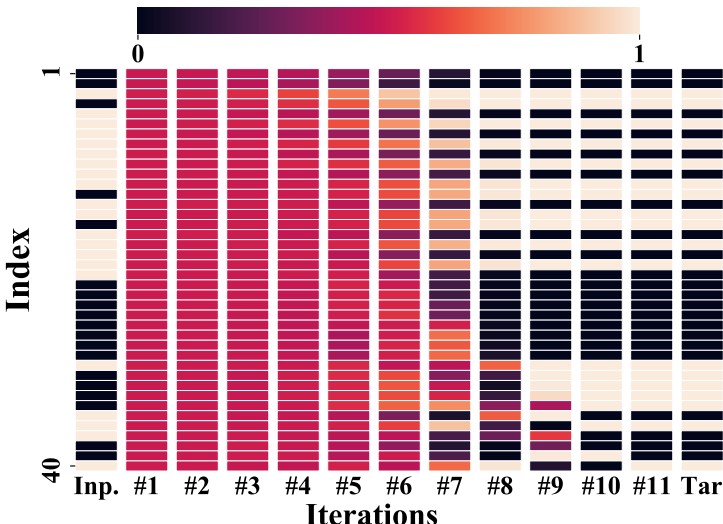

Figure 5: A recurrent model's output from each of 11 iterations on a 40-bit input string. Shown here is the confidence that there is a 1 at each index of the output. The first index is at the top for all vectors, the input is in the left-most column and the target is in the right-most column. The model used to produce this plot was trained with fewer iterations (10) on shorter input strings (32-bit).

### 4.2 Mazes

For maze solving, we train models on a training set composed of the easier small mazes, and we investigate the ability of networks to make the leap to larger, or harder, mazes at test time. In line with the findings above, we show two important behaviors. First, the recurrent models make the leap from small to large mazes better than feed-forward models. Second, when allowed to think deeper, the recurrent models exhibit even higher performance.

The networks employed here are fully convolutional and have 512 channels in the internal layers. In assessing the confidence of a given output, we average each pixel's classification confidence.

Table 2 shows that for a fixed effective depth, recurrent models always generalize to the hard mazes better than their feed-forward counterparts. Inspired by the upward trend in Table 2, we shift focus to deeper models. In Figure 6, we show that recurrent models can extrapolate to harder problems better than feed forward models. When trained on small mazes with 20 iterations (effective depth

Table 2: **The average accuracy (%) of models trained on small mazes and tested on large ones.** Over a range of effective depths, we see that recurrent models generalize to the harder mazes better than their feed-forward counterparts. Figures reflect averages over several trials ± one standard error.

| | Effective Depth | | | | |
| --- | --- | --- | --- | --- | --- |
| | 20 | 24 | 28 | 36 | 44 |
| Recurrent | $12.66 \pm 0.44$ | $14.02 \pm 0.39$ | $19.95 \pm 0.31$ | $22.96 \pm 1.03$ | $29.72 \pm 1.22$ |
| Feed-forward | $7.94 \pm 0.36$ | $12.43 \pm 0.50$ | $14.67 \pm 0.54$ | $17.71 \pm 0.36$ | $22.53 \pm 1.14$ |

of 84 layers), these networks can solve about half of large mazes. However, when allowed to think for longer, the recurrent models can correctly solve an even higher proportion of large mazes. In fact, models trained with 20 iterations can achieve upward of 70% accuracy on large mazes using 5 additional iterations at test time.

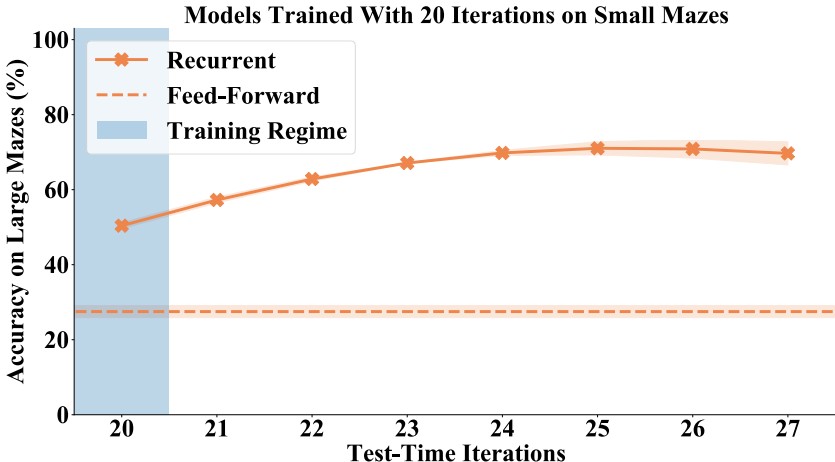

Figure 6: Generalizing from easy to hard mazes. We compare recurrent models to the best feed-forward models. The markers are at average values from several trials and the shaded regions indicate ± one standard error.

Maze solving is another task for which global information is needed. We investigate how dilated filters affect model performance. Dilation is a way of changing the receptive field without adding new parameters or more depth. Table 3 shows that dilations lead to slight improvements, however, the benefits of recurrence are still abundantly clear. Indeed, the difference in performance is even larger.

Table 3: **The average accuracy (%) of models with dilated filters trained on small mazes and tested on large ones.** Figures reflect averages over several trials ± one standard deviation.

| | Effective Depth | | | | |
| --- | --- | --- | --- | --- | --- |
| | 20 | 24 | 28 | 32 | 36 |
| Recurrent | $33.60 \pm 1.06$ | $40.49 \pm 1.63$ | $33.91 \pm 1.99$ | $40.56 \pm 4.34$ | $50.50 \pm 7.97$ |
| Feed-forward | $19.73 \pm 0.72$ | $21.59 \pm 0.22$ | $24.18 \pm 1.33$ | $25.82 \pm 0.10$ | $26.54 \pm 2.13$ |

#### 4.2.1 Iterative outputs

The recurrent maze solving networks also produce output at every iteration. Examining this output again leads to a remarkable conclusion: these recurrent models are narrowing in on the answer with each successive iteration. In Figure 7, it is clear from the output on iteration four that the network has found two routes emanating from the red square. Moving through the iterations, the model refines the output, increasing the confidence for pixels on the path and decreasing the others until finally at Iteration #7, the output matches the target. It is also interesting to observe here that the output is consistently correct for two iterations, after which a few pixels flip (Iteration #9).

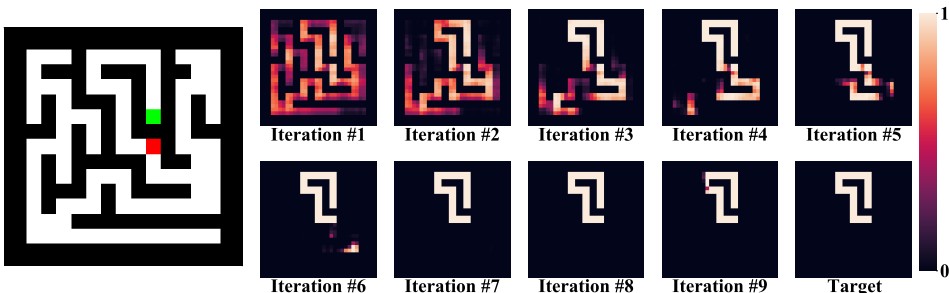

Figure 7: Input, target, and outputs from different iterations are shown to highlight the model's ability to think sequentially about mazes. We plot the model's confidence that each pixel belongs to the optimal path. This is a representative example from a model trained to solve small mazes in six iterations.

## 4.3 Chess puzzles

The third dataset comprises chess puzzles, or mid-game chess boards for which we seek the best next move. Unlike the other two datasets, the state of the art for chess playing algorithms is complex and has components that use algorithms like Monte Carlo tree search as well as neural network based elements for evaluating positions [Romstad et al., Silver et al., 2017]. The complicated nature of these systems provides some context for how difficult these puzzles are. What makes these puzzles particularly useful for us is that there is a predetermined best next move. A move is defined as an origin square, or the current location of the piece to be moved, and a destination square.[2] In order to generate target outputs for our models, we define a move as an $8 \times 8$ array with zeros everywhere except at the entries corresponding to the origin and destination squares which are ones.

We compare recurrent and feed-forward networks of effective depths from 84 layers to 100 layers that take $8 \times 8 \times 12$ arrays as input. These fully convolutional networks have 512 channels in the internal layers, and the output is $8 \times 8 \times 2$, corresponding to binary classification at each input pixel. During training, we use an average of cross-entropy losses at every pixel. When evaluating these models, however, we define the predicted move by the locations of the two highest confidence scores.

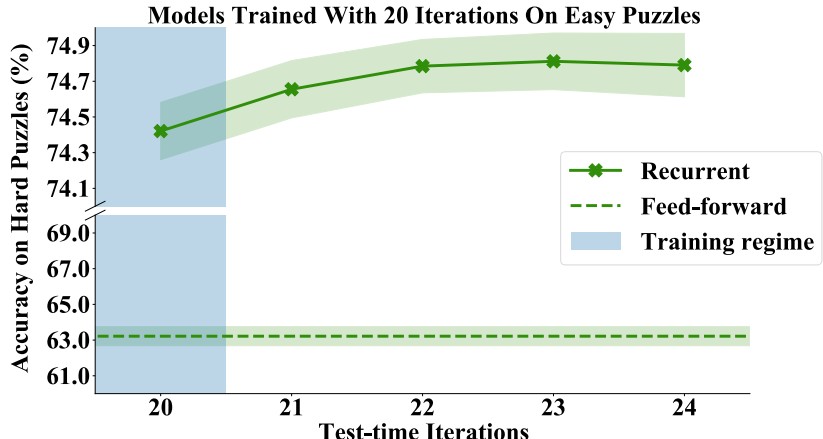

Figure 8: Generalizing from easy to hard chess puzzles. The ability of networks to solve harder puzzles than were used for training. We compare recurrent models to the best feed-forward models of comparable effective depth. The markers are at average values from several trials and the shaded region indicate $\pm$ one standard error.

---

[2]There are some cases, pawn promotions, where this information does not uniquely identify the move, and they are overlooked in this project.

Once again, we see that recurrent models can solve more chess puzzles than their feed forward counterparts. Furthermore, by thinking deeper at test time, recurrent models can perform even better. While the gains shown in Figure 8 are modest in comparison to the other two problem settings, the trend is clear – recurrent models can solve more puzzles with more iterations.

### 4.3.1 Iterative outputs

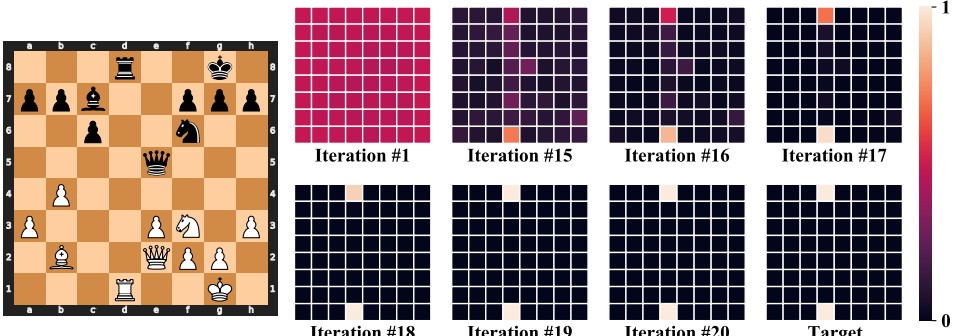

Figure 9: Input, target, and outputs form different iterations are shown to highlight the model's ability to think about the next move. We plot the model's confidence that each pixel is one of the two that define a move. In this example, black is to move next. For space consideration, iterations 2-14, which look like the first iteration, are left out of this plot. More examples are available in Appendix C.

While the outputs from the other two datasets show the similarity between the learned iterative process and known sum and search algorithms, extracting that insight on chess puzzles is much more difficult. Partly, this is because exploiting known search algorithms on such a large search space is hard to visualize. Nonetheless, the network's output, shown in Figure 9, tells a fascinating story. First, the Iteration #1 plot shows that after one iteration, we observe equal confidence at every location on the board. In the next frame, Iteration #15, it is clear that the model is considering moving the D8 rook to multiple squares, and it also considers the intuitive idea of using the E5 queen to place the white king in check on H2. With each successive iteration, the network becomes more confident – first that the rook is the correct piece to move, and then where to move it to.

## 5    Discussion

More than an answer, the results and conclusions in this paper are posing a question: Can learned models behave like classical algorithms in the way they generalize to harder or larger problems?

Our discussion of this question and our observations begins with delineating the limitations of our work. The first major limitation is that we do not propose a definitive answer. Rather, using representative cases, we demonstrate that recurrence can help neural models make the leap from easy training data to hard testing examples. A more subtle limitation of our work lies in how we split the data by difficulty. For prefix sum computation and for mazes, the classical algorithms approach to measuring problem complexity is tied to problem size, so in those settings we make intuitive easy/hard splits. With chess however, the issue is much more complex. Should puzzles with higher ratings require more memory or more computation? We carry out our work assuming the answer is yes, but we remain open minded to the possibility that the ratings assigned by Lichess may be weak surrogates for algorithmic complexity of each puzzle. In short, chess is an extremely difficult domain to analyze.

The observations above indicate that iterative models can learn processes that generalize beyond the training distribution with more iterations. One exciting use case is when the testing distribution is inaccessible, a setting where training on the harder distribution itself would be impossible. Many real life scenarios demand exactly this type of problem solving, from robots deployed in the real world after training in simulation to humans who spend a lot of time practicing on easy math problems only to spend years on difficult unsolved questions.

On a conceptual level, the recurrent model behavior we show is analogous to the human behavior of manipulating representations in working memory; in this analogy, the recurrent block performs the transformations and the activations it generates are the memory. It is not the goal of the experiments here to suggest that iterative models use mechanisms similar to those in a human brain. Nonetheless, it is exciting to see, even in a proof of concept setting, neural models that appear to deliberate on a problem until it is solved and can extend their abilities by thinking for longer.

This raises a questions that motivates future work. Can we build neural networks that can think for even longer? Is it feasible to have models whose performance only increases with added compute time? Humans who are given more than enough time to solve a maze, will not suddenly get it wrong after arriving at the right answer, perhaps this awareness of when to stop thinking can be built into networks like the ones we study here.

## 6    Conclusion

In this work, we demonstrate that neural networks are capable of solving sequential reasoning tasks and then extrapolating this knowledge to solve problems of greater complexity than they were trained on. These recurrent models are largely inspired by the classical theory of mind, in which the brain iteratively applies primitive strategies to solve complex problems over time [Baddeley, 2012]. Our models are recurrent versions of popular architectures and we acknowledge that variations to the model may be helpful. Thus, we leave an in depth investigation into other neural network designs for future work.

Interestingly, the resulting models excel at solving problems that are classically solved by hand-crafted algorithms; prefix sums are computed using reduction trees, mazes are classically solved by depth/breadth first search, and chess is solved by Monte-Carlo tree search. Even with the advances in machine learning that we have today, hand-crafted algorithms still play a role in state-of-the-art reasoning systems. A prominent example is AlphaZero, which plays board games using Monte-Carlo tree search algorithms assisted by a learned pruning function [Silver et al., 2017]. While moving away from this paradigm that includes hand-crafted elements is highly ambitious, this work suggests that it may be possible to train gameplay systems without building them on top of a hand-crafted tree search engine. In other words, it may be possible to machine-learn these algorithmic behaviors end-to-end.

## Acknowledgements

This project was supported by the ONR MURI program, AFOSR MURI program, the DARPA Young Faculty Award, and the National Science Foundation Division of Mathematical Sciences. Additional support was provided by Capital One Bank and JP Morgan Chase. Additionally, we thank Avrim Blum for thought-provoking and insightful conversations.

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
