# A Technical details

## A.1 Datasets

Details of the datasets we introduce are presented in this section. Specific details about generation as well as statistics from the resulting datasets are delineated for each one below. The datasets can all be downloaded from https://github.com/aks2203/easy-to-hard-data [Schwarzschild et al., 2021].

### A.1.1 Prefix sum data

Binary string inputs of length $n$ are generated by selecting a random integer in $[0, 2^n)$ and expressing its binary representation with $n$ digits. Datasets are produced by repeating this random process 10,000 times without replacement. Because the number of possible points increases exponentially as a function of $n$ and the size of the generated dataset is fixed, it is important to note that the dataset becomes sparser in its ambient hypercube as $n$ increases. Moreover, we are limited to datasets with binary strings of length $n > 13$ to avoid duplicate data points.

### A.1.2 Maze data

The maze data is generated using a depth first search algorithm. A grid is initialized with walls at every cell boundary. Then, using depth first search from a starting cell, every cell is visited at least once, removing walls along the path. The algorithm is available in the attached code. The resulting dataset has non-uniformly distributed path lengths. Also, this process does lead to duplicates, but fewer than $0.5\%$ of points are duplicated and so this is ignored in our study.

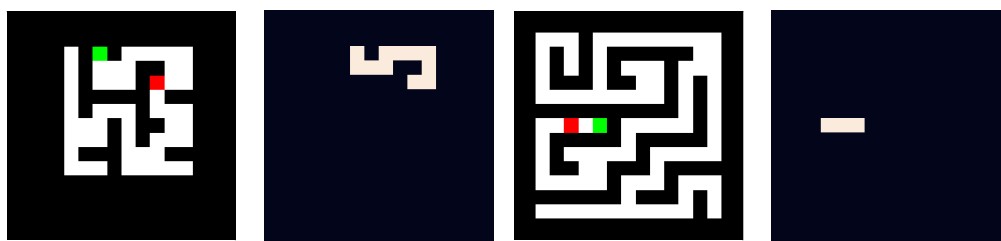

Figure 10: Example of small (left) and large(right) maze inputs and targets. The target is a binary classification label for each pixel indicating on/off the optimal path.

### A.1.3 Chess puzzle data

Starting position data from the Lichess database in FEN form are transformed into $8 \times 8 \times 12$ arrays indicating the position of each piece on the board (one channel per piece type and color) to be used as inputs. The first correct output moves in UCI move format are transformed into $8 \times 8$ binary masks showing the origin and destination positions for the optimal move and used as targets for the corresponding input. More specific details about these transformations can be viewed the provided code.

Lichess compiles puzzle data by analyzing 200,000,000 from their database of games between users using the Stockfish 12/13 NNUE chess engine at 40 meganodes. Though amalgamated puzzles do not come with username information, this information can ostensibly be gathered by playing through the puzzles on the Lichess website, where this information *is* available. Because of this, applying deep learning to this dataset could reveal some information about the playing style of Lichess users. However, upon creating accounts users agree to allow their game data to be public and be used specifically for investigative purposes.

The puzzle data is released by Lichess for public use under the Creative Commons CC0 license.

## A.2 Architectural details

Model files are available in the linked code repository, their details are as follows.

The feed-forward prefix sum models are fully convolutional models that take in $n \times 1$ arrays. The first layer is a one-dimensional convolution with a three entry wide kernel that strides by one entry with padding by one on either end on the input. The output of this first convolution has 120 channels of the same shape as the input. The next parts of the networks are residual blocks made up of four layers that are identical to the first layer with skip connections every two layers. After the residual blocks, there are three similar convolutional layers that output 60, 30, and two channels, respectively. For a network of depth $d$, there are $(d-4)/4$ residual blocks. The recurrent models are identical, except that all residual blocks share weights.

The feed-forward maze solving models are fully convolutional models that take in $n \times n \times 3$ arrays. The first layer is a two-dimensional convolution with a $3 \times 3$ kernel that strides by one entry and pads by one unit in each direction. The output of this first convolution has 128 channels of the same shape as the input. As above, the next parts of the networks are residual blocks made up of four layers that are identical to the first layer with skip connections every two layers. After the residual blocks, there are three similar convolutional layers that output 32, 8, and two channels, respectively. For a network of depth $d$, there are $(d-4)/4$ residual blocks. The recurrent models are identical, except that all residual blocks share weights.

For experiments with dilated filters, the only changes made are to the dilation of the convolutional filters and to the padding of every convolution and the values are set to maintain the output dimenssion of each layer.

The chess playing models are the same as the maze models except that the first layer takes $8 \times 8 \times 12$ inputs and outputs 512 channels.

None of the models used in this project have batch normalization or bias terms.

### A.3   Training hyperparameters

The training details and hyperparameters are outlined below.

- Prefix sum training is unstable, thus we only save models that show 100% training accuracy at the end of training.
- Data augmentation: Binary string inputs to prefix sum networks are approximately normalized by subtracting 0.5 from every element in the string in order to aid in training stability. Mazes are padded to be $32 \times 32$ pixels.
- Optimizer: All prefix sum networks are trained using the Adam optimizer with a weight decay factor of 2e-4. Because of training instability, we also apply gradient clipping at magnitude 1.0. The maze solving networks are trained with stochastic gradient descent with a weight decay factor of 2e-4 and momentum coefficient of 0.9. Chess networks are trained using stochastic gradient descent with a weight decay factor of 2e-4 and momentum coefficient of 0.9.
- Epochs: Prefix sum networks are trained to convergence with 500 epochs. Maze models are trained for 200 epochs. Chess networks are all trained to convergence with 140 epochs.
- Learning rate and decay schedule and type. All prefix sum networks are trained using an exponential warm-up schedule applied over 10 epochs. Initial learning rate (post-warmup) is set at 0.001 and is subsequently halved at epochs 100, 200, and 300. Maze solving networks also use warm-up with a period of 5 epochs after which the learning rate is 0.001. The learning rate further decays by a factor of ten at epoch 175. Chess networks are trained with an exponential warm-up schedule applied over 3 epochs. Initial learning rate (post-warmup) is set to 0.1 and dropped by a factor of ten at epochs 100 and 110.
- Batch size: For training prefix sum models, we use batches of 150 binary strings. When training maze networks we use batches of 50 mazes. For chess models, we train with batches of 300 puzzles.

### A.4   Compute resources

All of our experiments were done on Nvidia GeForce RTX 2080Ti GPUs. Prefix sum models train in less than one hour on a single GPU, whereas maze solving models require approximately seven hours on a single GPU. The chess networks were trained in about 24 hours on four GPUs.

In total, the prefix sum training required to generate the results presented in this paper takes approximately 30 GPU hours. The maze models, in total, require one GPU week. And the chess networks took 3 GPU weeks.

The data pre-processing as well as the testing of all models for every experiment can done comparatively quickly, in several hours.

## B    Further insights

We present several additional findings from our experiments.

### B.1    Deeper feed-forward models

Significantly deeper feed forward models cannot generalize from easy to hard as well as recurrent ones. For example, we extend the experiments in Figure 4 to feed-forward models with depths 132, 264, and 528. These models all fit the training data (32-bits) well, but the best performance on 40-bit data is 53% and on 44-bit data is 27% – neither is attained by the deepest models lending confidence that the performance would not increase with even more depth.

### B.2    In-distribution tests

For in-distribution test accuracy (on small/easy cases) on prefix sums, both classes of models achieve >99% and on mazes, both achieve >97% (using the same number of iterations as used during training). When we apply more iterations, we see only slight drops for in-distribution test accuracy (tenths of a percent), but no improvement. On chess puzzles, the in-distribution test accuracies are 92% for the recurrent models and 84% for the feed-forward networks (though there is a gap in performance, it is smaller than the gap observed for out-of-distribution testing).

We also train and test on the harder datasets to measure in-distribution accuracy. Here, we find there is little difference between recurrent and feed-forward models. As examples, both classes of models for prefix sums and mazes achieve essentially 100% on unseen data. Specifically, both recurrent and feed-forward models achieve >99% on prefix sums and >97% on mazes.

Finally, we train models on larger instances and test them on smaller cases, and we find that they can solve smaller examples in fewer iterations than were used at training. For example, when models are trained to compute prefix sums on 44-bit data in 10 iterations, they can get 100% of the 16-bit test examples correct after only 4 iterations.

### B.3    Prefix sum experiments on other datasets

### B.4    Dilated filters

The receptive field can be increased without adding parameters or depth by dilating the convolutional filters. When we use dilated filters to compute prefix sums, we find that we can fit the training data with $N$-bit sequences with fewer than $N$ layers – a behaviour that is not possible with non-dilated convolutions. This suggests that the algorithm learned by our models is informed by the receptive field. In other words, since the final entry in a prefix sum does require global information (and therefore, in order for a neural network to compute the final entry it needs a complete receptive field), the use of extra iterations on longer sequences meshes with the algorithmic analysis. When we contextualize our models by analyzing the way they scale, it is reasonable to find that networks with non-dilated convolutions scale linearly with the input length. This also motivates future work to study whether neural networks can be designed to learn an algorithm that scales with the square root of the problem size, or even logarithmically.

### B.5    Even harder chess puzzles

## C    Visualizations

Additional visualizations of intermediate outputs, along with input and target examples from all three datasets are presented below.

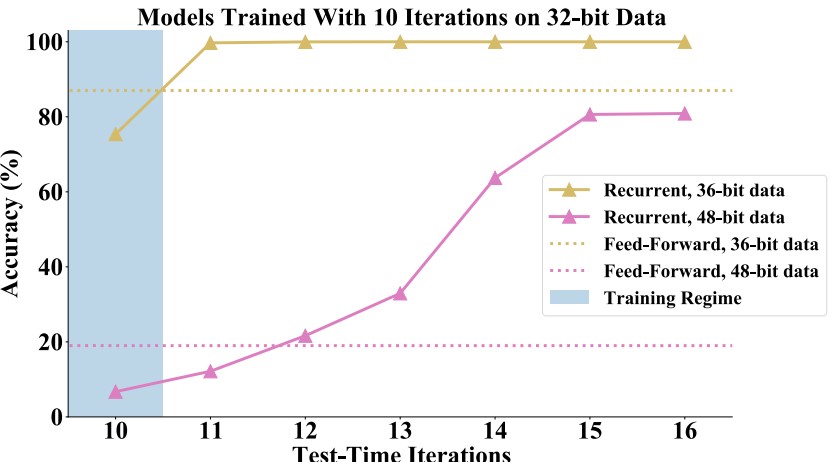

Figure 11: Generalizing from easy to hard prefix sums. The ability of networks to compute prefix sums on two additional test sets with longer input strings than were used for training (accuracy on 36-bit inputs in yellow and on 48-bit inputs in pink). We compare recurrent models to the best feed-forward models of comparable effective depth.

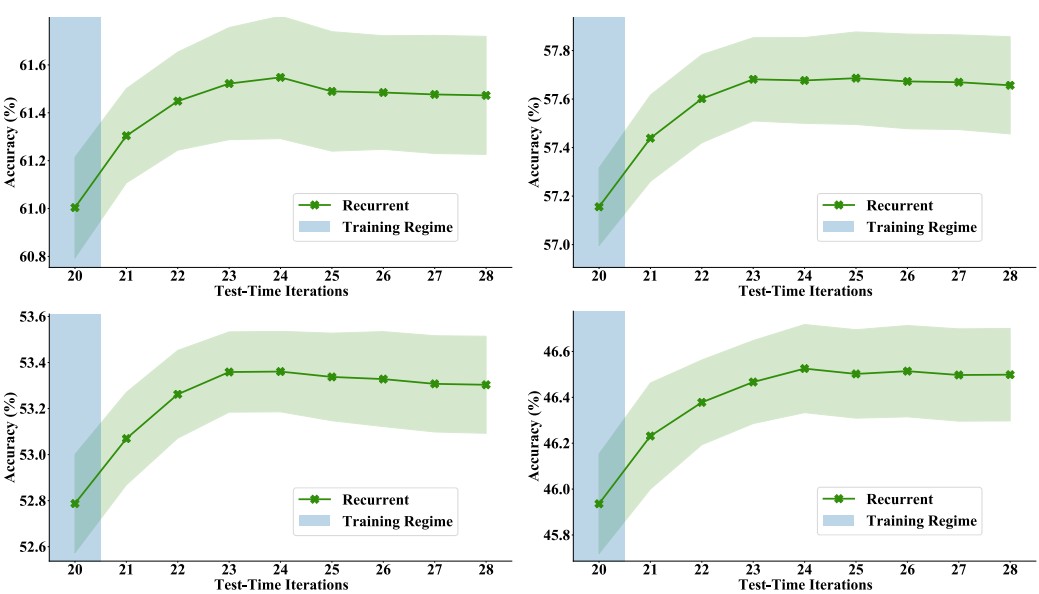

Figure 12: Generalizing from easy to hard chess puzzles. The ability of networks to solve harder puzzles than were used for training. We trained models on the first 600,000 puzzles and we show their performance with extra iterations on puzzles with index 800,000 to 850,000 (top left), 850,000 to 900,000 (top right), 900,000 to 950,000 (bottom left), and 100,000 to 150,000 (bottom right).

## C.1 Prefix sums

We show a recurrent model's output from each of 11 iterations on 40-bit input strings. Shown below is the confidence that there is a 1 at each index of the output. The first index is at the top for all vectors, the input is in the left-most column and the target is in the right-most column. The model used to produce these plots was trained with fewer iterations (10) on shorter input strings (32-bit).

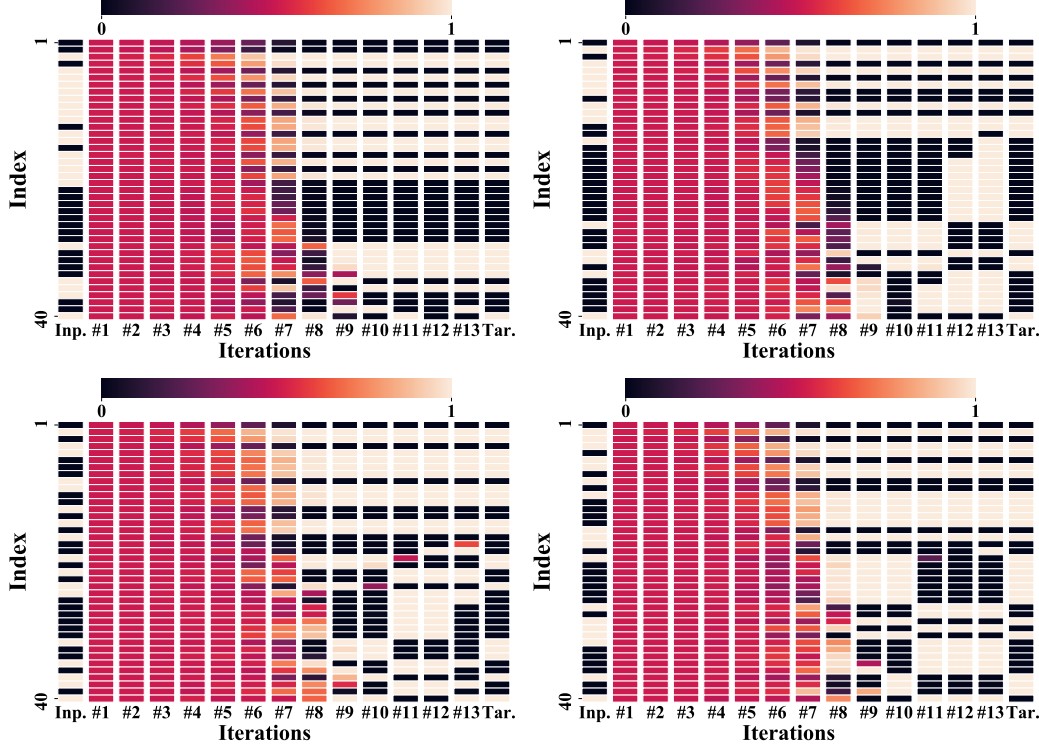

Figure 13: Prefix sum intermediate outputs.

## C.2 Mazes

We show inputs, targets, and outputs form different iterations to highlight the model's ability to think sequentially about mazes. We plot the model's confidence that each pixel belongs to the optimal path. Below are several representative examples from a model trained to solve small mazes in six iterations.

## C.3 Chess puzzles

We show inputs, targets, and outputs form different iterations to highlight the model's ability to think about the next move. Below, we plot the model's confidence that each pixel is one of the two that define a move.

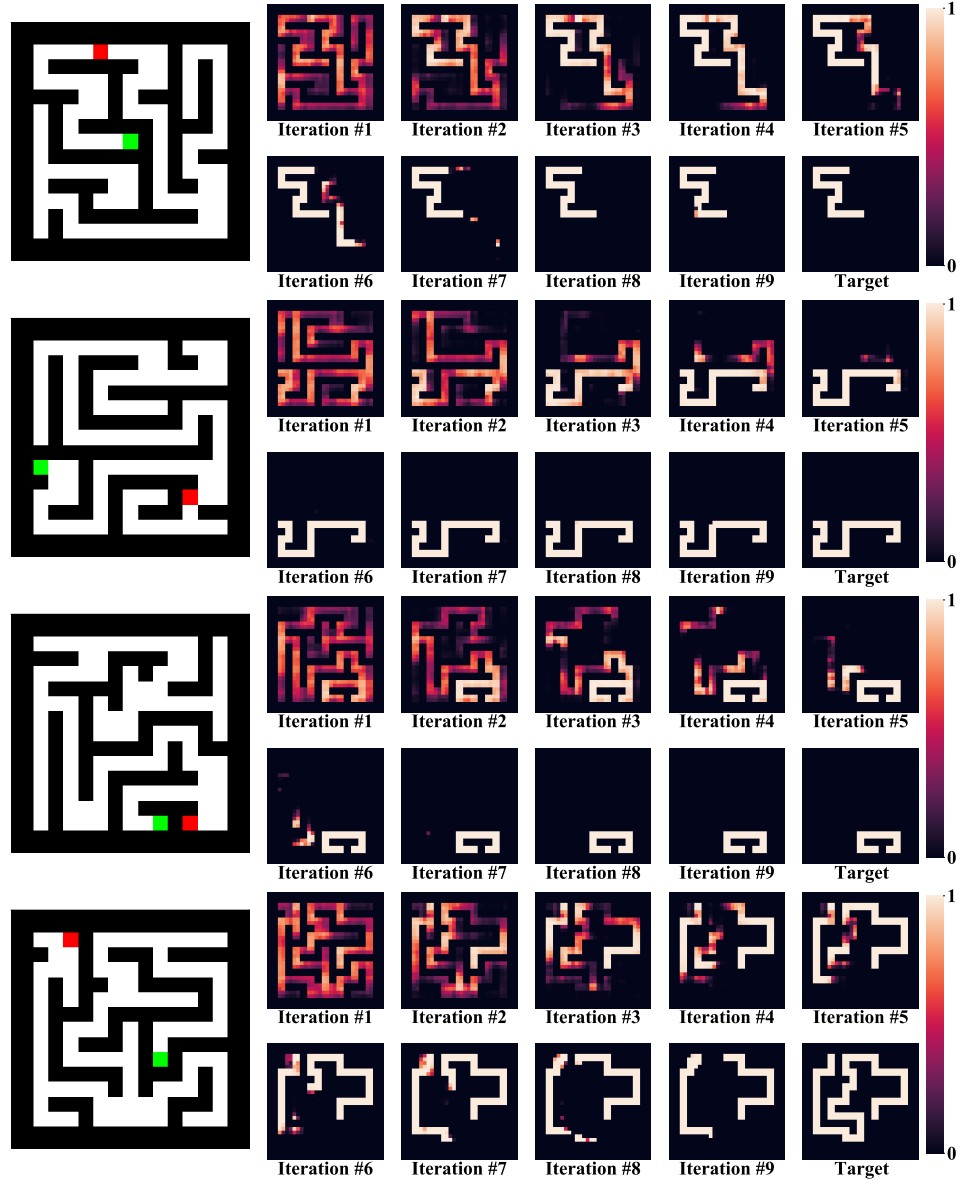

Figure 14: Maze model intermediate outputs.

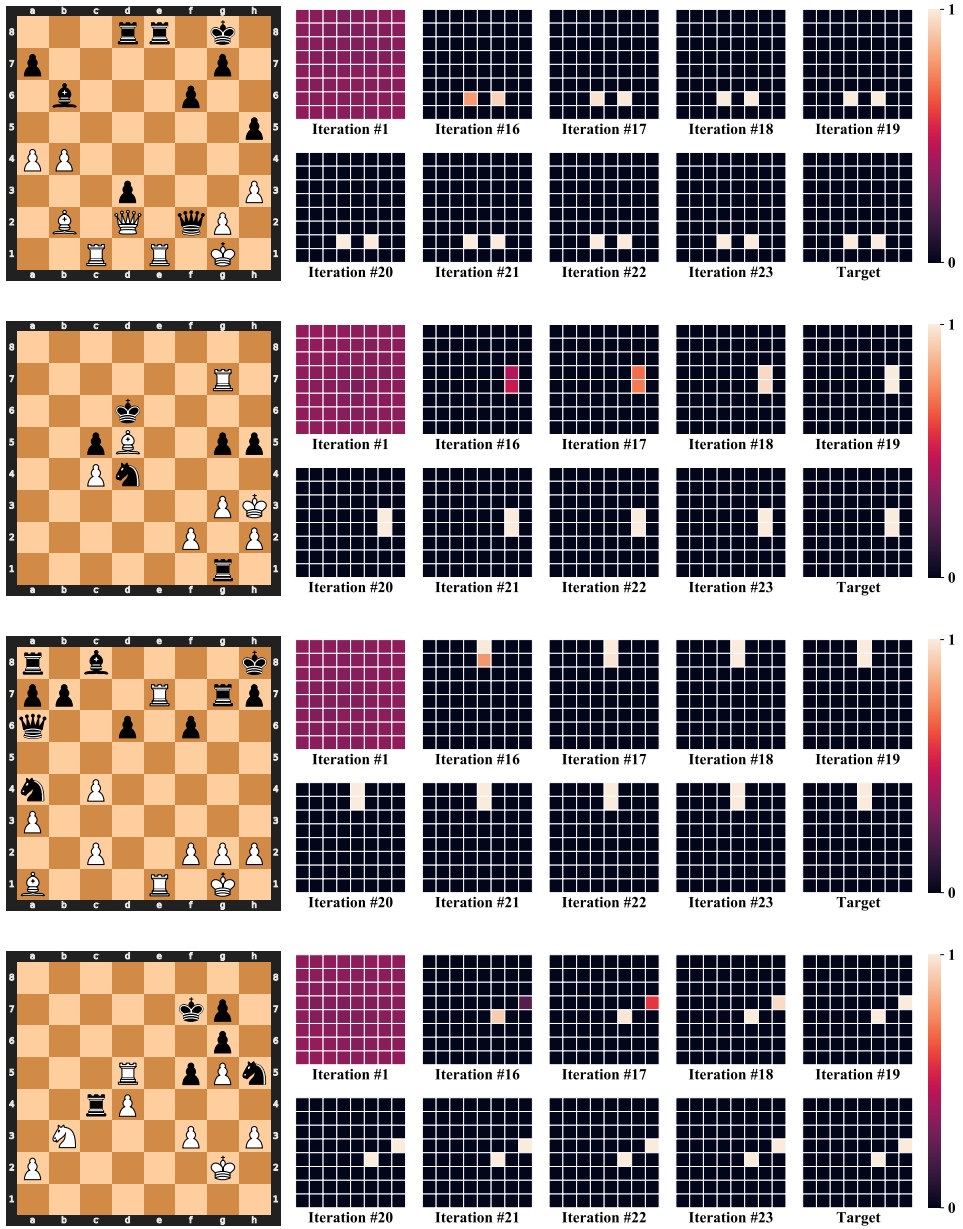

Figure 15: Chess model intermediate outputs.