# OpenReview forum: "Can You Learn an Algorithm?  Generalizing from Easy to Hard Problems with Recurrent Networks"
_NeurIPS.cc/2021/Conference — NeurIPS 2021 Poster_

### Official Review · Reviewer_iKnp · 2021-06-29

**Rating:** 4
**Confidence:** 5

**Summary:**

The paper demonstrates generalization on 3 tasks: prefix sum, maze and chess.
A fully convolutional net with a repeated recurrent block was able to generalize to larger input or harder cases without retraining.
A higher amount of recurrent iterations was helpful on the harder cases.

**Limitations And Societal Impact:**

No negative societal impact.

**Main Review:**

The paper correctly positions itself as a motivational paper, presenting important questions instead of a recipe for generalization.
The main weakness is that the paper is not aware of the solutions in the previous work:
1) "Neural GPUs Learn Algorithms" ... Neural GPUs generalized to longer inputs.
2) "Value Iteration Networks" ... explained that 2D mazes can be solved perfectly with convolution and max-pooling.
3) "Unbiasing Truncated Backpropagation Through Time" ... the generalization to longer thinking time is related to truncated backprop.

Pros:
- The paper is well written. Nicely readable.
- Generalization to larger inputs and longer thinking time is an important problem.
- The experiments are well done. The comparison to the feedforward nets is informative.
- Nice visualizations of the iterative outputs.
- Nice trick: Allowing to do more iterations and taking the output with the highest confidence.

Cons:
- The paper does not provide an advice for generalization to longer thinking time. It would be helpful to mention the problematic parts of the training: truncated backprop, distribution shift without providing an ability to adapt, ...
- The net has not achieved 99% accuracy on the 2D mazes. Value Iteration Networks worked better.
It would be helpful to report also the accuracy on the training set.

Minor typos:
- Line 138: "as" should be "is".
- Line 214: "form" should be "from".
- Figure 5: The input does not seem to correspond to the target.

**Time Spent Reviewing:**

4

---

> ### Author Response · Authors · 2021-08-10
> **Thank you for the review**
>
> We have updated our draft to include the following contextualization of our work among the references you mentioned.  All of these sources are worth citing.
>
> The Value Iteration Network paper does not consider the extrapolation of a VIN from small problems to large problems (which is the primary focus of our paper). Nonetheless, we find it interesting to experiment with whether such a task-specific architecture can achieve this extrapolation, and we have cited VIN in our updated draft.  Prompted by your review, we have now run experiments with VIN (using code provided by the VIN authors), and we have found that the method fails to generalize from small to large mazes (our data), achieving less than 20% `success` (by the VIN definition of “correct”, which requires a path from start to finish, but does not require an optimal path as required by the stricter definition in our paper) on the large mazes when trained to 99% training accuracy on small mazes.  In contrast, our proposed framework achieves 70% accuracy on large mazes after training only on small mazes - and this 70% is calculated using our more restrictive definition of ‘success’ that requires a path to be optimal.
>
> The reviewer claimed that “The net has not achieved 99% accuracy on the 2D mazes.  Value Iteration Networks worked better.”  You are correct that VIN has been successful when tested on problems of the same size on which it was trained.  However this does not pertain to the issue of generalizing to larger problems.  Moreover, our in-domain accuracy on mazes is also >97%, and we will add these numbers to our paper for comparison.
>
> The VIN network also benefits from a number of hand-crafted rules; the network is crafted so that it always generates a continuous path.  It also has a hand-crafted termination rule that stops the algorithm when the correct goal cell is reached.  Our work differs from VIN in that it (1) can learn behaviors that scale to larger problems, (2) uses problem-agnostic architectures with no hand-crafted updates, and (3) uses automatic stopping rules without task specific hand-crafting.  In fact, we use the same architecture and stopping rules for all problems considered in our paper (with the exception of swapping 2D convolutions for 1D convolutions for the prefix sums).
>
> Thank you for bringing ARTBP to our attention, and we have now cited it in our draft.  While ARTBP only backpropagates through small subsequences, it still trains (computes forward passes) on the full sequences using truncated backprop for efficiency, and the issue of generalizing from small training samples to large testing samples is not considered.  Moreover, our proposed models do not take in data as sequences.  Nonetheless, this is an interesting work, and we have addressed it in our updated version.
>
> We have carefully read your references on the neural GPU.  This is an interesting early work studying whether sequence-to-sequence models can learn to reproduce bit sequences generated by boolean formulas.  While the focus of the paper is narrower than ours, it is certainly related work, and we have added a discussion about it to our literature survey.
>
> Thank you for suggesting additional edits, and we have made the associated corrections.

---

> > ### Comment · Reviewer_iKnp · 2021-08-23
> > **Response to authors**
> >
> > Thank you for your detailed reply.
> >
> > I still feel that the "Can You Learn an Algorithm?" question was already experimentally addressed 5 years ago, by Neural GPUs and other works. I would welcome theoretical guarantees or new algorithms.

---

> > > ### Author Response · Authors · 2021-08-31
> > > **Author Response**
> > >
> > > Neural GPUs are designed explicitly for sequential inputs. In fact, the results in (Kaiser & Sutskever 2016) are only on binary bit strings. Our results are therefore a valuable exploration into algorithm learning work.
> > >
> > > Furthermore, the structure of neural GPUs limits their generalization capabilities to longer sequences. There is no mechanism by which they can think deeper on harder tasks of the same size, which makes them completely inapplicable for problems like chess.

---

### Official Review · Reviewer_idGE · 2021-07-06

**Rating:** 5
**Confidence:** 4

**Summary:**

This paper proposes the idea of learning an iterative refinement function within the deep learning framework using recurrence where training is done over a fixed number of iterations, but testing can be performed over an arbitrary number of iterations to let the learned model find a solution in case it needs to “think longer”. The authors have experimentally shown that the learned models can indeed solve more out-of-distribution problems if they're let run longer for three Prefix-sum, Maze and Chess Puzzles problems. They have also shown for the same out-of-distribution problems, a feed-forward model of the same effective depth performs poorly compared to their proposed recurrent model. Based on these results, the authors have claimed that the recurrent model has superior out-of-distribution generalization compared to its feed-forward counterpart. These out-of-distribution datapoints are specifically characterized as “larger” and “harder” throughout the paper, which in turn has paved the way to make the main claim of paper: RNNs can be used to train on easy problems and then generalize over the hard ones at the test time by letting them run longer.

**Ethical Concerns:**

There are no ethical issues with this work.

**Limitations And Societal Impact:**

As for limitations of their work, the authors have gone to a great deal explaining them in Sec 5, which is all fair and well-explained. Although, as I explained above some of the main claims of the paper seem to be unsubstantiated or at best very specific to the experiments done. So in that regard, I'd strongly recommend authors to either provide more proof for those claims OR revise them by explicitly discussing their limitations.

As for societal impact, I agree with the authors that their work does not particularly pose any negative impact beyond that of the general field of AI.

**Main Review:**

Originality:
The idea of injecting the inductive bias of “iterative refinements through recurrence” into deep learning (or as the authors have put it “thinking longer”) and then arbitrary iterate over it at the test time is indeed an interesting and quite a useful idea, but it’s hardly a new one. In fact, people have incorporated the exact same idea in neural SAT and CSP solving. Here are two samples:

-  Selsam, D., Lamm, M., Bünz, B., Liang, P., de Moura, L., & Dill, D. L. (2018). Learning a SAT solver from single-bit supervision. arXiv preprint arXiv:1802.03685.
-  Amizadeh, S., Matusevych, S., & Weimer, M. (2019). PDP: A general neural framework for learning constraint satisfaction solvers. arXiv preprint arXiv:1903.01969.


Significance:
While the experimental results are quite interesting (especially the behavior of the recurrent models with iterative outputs), the paper would have made a significant contribution on the top of the original idea by either providing a theoretical characterization of the sub-class of problems where such inductive bias is applicable OR implementing the idea in more practical applications such as Computer Vision where ConvNets are typically trained on a narrow distribution of image dimensions and generalization to completely different size images can be an issue. However, this work has provided neither of those two which makes the overall contribution somewhat marginal given that the main idea was used before in the literature.


Clarity:
The paper is very well-written with clear statements, explanations and enough details.


Quality:
The overall quality of the paper is acceptable; however, there are a few significant issues that overshadow some of the main claims of the paper:

-  Size vs hardness:
Throughout the paper, the authors seem to confuse the notion of the “size” of a problem with its “hardness”. These are generally two different things. In SAT solving, for example, the size of a problem is determined by its number of variables; whereas, its difficulty is reflected by the ratio of the number of its clauses over the number of its variables. That is, a larger problem is not necessarily a harder one, and therefore scale-invariant algorithms are *not* necessarily guaranteed to generalize to “harder” problems. Now, for prefix-sum and maze problems, it seems to me that the test dataset contains “larger” problems and *not* the harder ones, due to the recursive sub-structure of those problems. For the chess puzzles, on the other hand, it seems the test data indeed contain harder examples; however, as the authors have also noticed, the gain is “modest”. Therefore, overall it seems one of the main claims of the paper that the proposed framework “generalizes from easy to hard” is not supported by the experiments.

Even from the scale-invariance aspect, it’s hard to say how scale-invariant (if at all) the learned recurrent models are, given that the out-of-distribution results are only reported for 1 (Maze & Chess) or 2 (Prefix Sum) datapoints (i.e. distributions). Ideally, it’d be interesting to see (A) whether the learned model consistently scales up with larger and larger problems and if so, (B) at what rate the number of required iterations to solve a problem increases as the problem size grows. The latter would, in effect, show the effective complexity of the learned algorithm.

-  Generalization Comparison:
“In other words, recurrent models trained with relatively few iterations generalize well to harder data while similar and even much larger feed-forward networks fail to generalize in the same scenario.”
Generally speaking, the hypothesis space of a feed-forward network is a superset of that of the recurrent model of the same effective depth. This means that in theory, given enough data and a good regularization scheme, the feed-forward model should be able to learn the same (if not better) function as that of the (rollout) recurrent model, unless it overfits of course (which seems to be a probable factor in this case, due to the over-parameterization of the feed-forward model compared to the recurrent model). In that sense, it’s not clear how much of the poor “out-of-distribution” (OOD) generalization of the feedforward model is attributable to the vanilla overfitting vs. an actual inability to generalize to OOD examples. To claim that it’s all the latter and NOT the former, it’d be necessary to report the results of “in-distribution” generalization for each model and experiment. Only if those numbers are close for the two models, one can claim that the observed gap between the two models on OOD examples is fully attributable to the superior OOD generalization of the recurrent model. In other words, it’s hard to make claims about the relative OOD generalization powers of two models when their relative in-distribution generalization powers are still in shadows. Therefore, I’d strongly recommend the authors to include the in-distribution test accuracies as well for all the experiments.


Vision:
Lastly, in final conclusions of the paper, the authors have stated: “In other words, it may be possible to machine-learn these algorithmic behaviors end-to-end.” While this might be an interesting theoretical exercise, I’m not sure it’s the right vision from the practical perspective. In fact, this goes against the main intuition of neuro-symbolic methods where symbolic algorithms are incorporated within the deep learning framework in order to exponentially reduce the amount of training data and time needed for learning anything useful. Aside from humongous amount training data one would need to learn such algorithms “end-to-end”, at the end of the day, there’s still no correctness and scale-invariance guarantees for a purely statistical model.

**Time Spent Reviewing:**

6 hrs

---

> ### Author Response · Authors · 2021-08-10
> **Thank you for the detailed input**
>
> We have updated our draft to include a much-needed discussion of the referenced papers. We find Selsam’s results to be quite interesting and to have conceptual similarities to our approach.  While he does test MPNNs on harder data with extra message passing iterations, the results are specific to constraint satisfaction problems and focus quite strongly on message passing updates; They develop a problem-specific architecture with enforcement of negation and permutation invariance, and use iterations based on a hand-crafted message-passing process rather than using generic recurrent blocks.  They also use a specialized training dataset consisting of pairs of satisfiable and unsatisfiable problems that differ only by the negation of one literal.  In contrast, the models we study have a strikingly simple and generic convolutional architecture that does not require rule-based connectivity between nodes. Moreover, given how similar the architecture is to image classifiers, we agree that future work could indeed lead to exciting results such as improved generalization for computer vision, as was aptly suggested.
>
> Larger vs. Harder: Thank you for pointing out this ambiguity in our writing.  We have updated our draft to use “large” vs “small” instances whenever applicable to make the paper more clear.  We have found in experiments that for in-domain settings, larger problem instances require more depth (or iterations) to solve.
>
> Scale invariance: We kindly point you to the appendix where two additional prefix sum datasets and four additional chess datasets are used to demonstrate generalization to harder problems. We have updated the paper to reflect that we study the leap to several difficulties.
>
> Generalization: In most cases, the results for in-distribution generalization show that feed-forward and recurrent models of the same effective depth are empirically equivalent (numbers added to the appendix), but recurrent models exhibit better OOD generalization. We agree that the feed-forward model could learn the same parameters in each block. However, we would point out that the hypothesis space of feed-forward models of a specific depth is not a superset containing the hypothesis space of recurrent models that run for different numbers of iterations depending on the input.  More importantly, feed-forward models in general do not learn the same parameters in each layer and have no natural way to extend their compute, as opposed to recurrent models, for which the number of iterations can be expanded quite naturally.  In this regard, we claim that the difference in OOD generalization may be a fundamental limitation of the feed-forward architecture.
>
> Vision: In problem settings where effective algorithms already exist, it can be very useful to incorporate them into the training process. Though prefix sums and mazes are both easily solved by classical algorithms, they are presented here simply as a laboratory for experimentation. In practice, a practitioner might not know the invariances in their problem, and may not be able to craft an efficient algorithm to solve it.  Our work highlights the ability of networks to learn the rules and algorithms end-to-end without hand-crafting or low-level supervision.

---

> > ### Comment · Reviewer_idGE · 2021-08-22
> > **Reviewer idGE Rebuttal**
> >
> > Thank you for your responses to my questions and concerns. Nevertheless, I'm still not convinced by some of your answers.
> >
> > **Originality/previous work**: I do understand that both references I mentioned above have applied their methods on different classes of problems than your experiments. However, despite your claim of being more "generic" and less "hand-crafted", those frameworks are in fact more generic than your proposed model for two main reasons. (1) Those frameworks tackle the general class of SAT and CSP problems where many problems can naturally be reduced to; in your case, however, you have not identified what class of problems your recurrent ConvNet can be applied to other than the three examples you've used for your experiments. (2) More importantly, all these iterative search strategies (including yours and the previous work) boil down to a notion of local, *generalized message passing* where the local information is passed along iteratively until a global solution is achieved. In your framework, the message passing is carried out by the iterative convolution operation. In that sense, your convolutional framework is indeed a *special* case of the generalized message passing on graphs where the underlying graph of the problem has a grid-like structure (as opposed to having an arbitrary structure) like the three problems you've presented in the paper.
> >
> > **Larger vs. Harder**: Thank you for addressing the confusion, but this is beyond confusion. You have made a recurring claim throughout the paper (including the title) that your framework is capable of generalizing to "harder" instances of problems, which has a very specific meaning from the computational complexity perspective. And yet, nowhere in the paper that claim is proven, theoretically or experimentally. The closest instance to proving such claim is the chess case, but the results for that experiment are not particularly convincing. In the absence of such proof, you should completely remove that claim from the paper and its title and restrict the scope of contribution to studying scale-invariance properties of the proposed architecture.
> >
> > **Generalization**: I completely agree with the authors that iterative induction bias is quite useful in certain classes of problems and we should use it whenever possible as it reduces the number of model parameters while improving the generalization. There's no doubt in that general statement. But what I'm wondering here is whether those reported numbers in your specific experiments show that claim as opposed to being tainted by vanilla overfitting. And the concern is valid, as your feedforward baseline has as "number of iterations" times parameters as your recurrent model, hence quite prone to overfitting. Based on that, I'd like to know what exact numbers have you observed for in-distribution generalization of both feedforward and recurrent models. Can you please upload a new appendix containing them or at least report them here?

---

> > > ### Author Response · Authors · 2021-08-31
> > > **Author Response**
> > >
> > > (1) From the following in-distribution generalization results, we see that overfitting is not the culprit here -- feed forward nets have not simply overfit to training data relative to recurrent networks as a result of their greater parameter count. In-distribution test accuracy for prefix sums was 100 percent for recurrent networks and 99.63 for the feed forward models. For mazes the in-distribution accuracies were 97.38 for recurrent networks and 97.40 for feed-forward models. For chess the in-distribution test accuracies were 91.685 for the recurrent models and 83.75 for the feed-forward networks (though there is a gap in performance, it is smaller than the gap observed for out of distribution testing).
> > > (2) We agree SAT solvers and MPNNs are technically generic in that a very wide range of problems can be rephrased for those methods. However, we believe there is also a very wide range of problems on which SAT-solvers and MPNNs are impractical.  We misunderstood your earlier point, but we do not think that our work on CNNs should be written off as a special case of message-passing neural networks or SAT-solvers.
> > > (3)  We believe that “harder” can mean different things both colloquially and in various academic disciplines. We aim to use the word in a manner that reflects colloquial and neuroscientific usage, where difficulty is often measured by latency, a metric which correlates with time complexity.  Thank you for pointing out that our usage is confusing to those in theoretical computer science -- we have updated our draft by replacing the word “harder” with “larger” and by clarifying our notion of difficulty.

---

### Official Review · Reviewer_XL39 · 2021-07-09

**Rating:** 8
**Confidence:** 4

**Summary:**

This paper describes a method that allows models to extrapolate to larger problems than those they are trained on, a known limitation of many neural network architectures. It proposes a network with an encoder, a decoder and a recurrent unit (a 4 layer residual convolution network) that is iterated through several times during training, the output of an iteration being used as the input to the next. This amounts to a deep ResNet with shared middle layers.

The authors show that models trained with this architecture can generalize to larger problems by increasing the number of iterations of the recurrent unit during inference. This achieves better accuracy than feed-forward ResNets of equivalent depth.

Three problems are considered: parity integration of a binary string, path finding in random mazes (encoded as binary images), and solving chess problems (generated from Lichess). The models are shown to generalize to longer strings, larger mazes, and, to a lesser extent, harder problems (measured by their Elo ranking).



**Limitations And Societal Impact:**

They have.

**Main Review:**

**Originality**
The architecture proposed by the paper is inspired by the Adaptive Computation Time model (Graves 2017), and its successors. The authors provide references to this line of research. The building blocks are standard residual convolution networks. The problems studied have been considered by previous authors. The originality of the paper lies in leveraging the recurrent architecture to run more steps at inference, and generalize to larger problems.

There has been a lot of research on generalization to longer sentences in sequence to sequence models. Tree-LSTM (Arashahi, 2019) and attention mechanisms (Dubois, 2019) have been proposed, among others. This might be worth adding to the related work.

**Quality**
A very good paper, the methodology and the datasets considered are clearly exposed and the experiments and conclusions are very compelling. A number of additions might make it even better.

1- The terms "easy" and "hard" are a little misleading. What seems to be demonstrated in the case of parity integration and mazes is extrapolation to **larger** problems (in terms of description length of their input and output). This is not the case for chess, and it might be a reason why the method is less efficient. It would be good to clarify this definition of "hard problem" in the introduction (or use a different word).

2- For parity integration, the base operations over a long sequence uses no more memory or computation than those of a short one: every new output digit is a XOR between the previous output and the next token. It would be interesting to try a binary string problem where memory/computation increases with sequence length (e.g. computing square roots or any function where many or all previous input or output are needed at each step.

3- At inference, it is shown that iterating improves accuracy on larger problems. What happens as one goes on iterating? Does accuracy saturate or does it ends up decreasing? If the iteration curves saturate, can a stop condition be derived, either from the confidence map, or successive iteration accuracies? Such a decision rule would turn the experimental technique described in the paper into a  solver for the problem considered. Proving that the solution stabilizes as the model iterates is an interesting theoretical result, because it makes inference a fixed point process.

4- What about problems of the same size as those trained on? In the maze case, for a given dimension, one expects problem difficulty to be correlated with the length of the solution path, so there will be easy and hard problems inside the test set. Can iterating at inference improve test accuracy beyond training accuracy, over the same distribution? (This would be extremely interesting!)

5- What about smaller problems?  Models trained on large problems often badly generalize to simple cases. Can the model "dumb down" by computing more?

6- It is mentioned that parity integration is difficult to train. Could normalization help? More generally, a few ablation/architecture experiments on the shared parameters (less layers/lower dimension in the basic iteration blocks, normalization, maybe even a visual transformer layer) would be very welcome, and strengthen the paper.

**Clarity**
Clearly written. Providing the source code is a great help. A sketch of the architecture (for the mazes, say), explaining the difference between the training and inference computation, would be very helpful: this is where the novelty lies. The chess experiments are much less compelling than the others, and as you note in the discussion, more difficult to interpret. It might be a good idea to move some of them in the supplementary, and improve the discussion of other cases in the main paper.

In the appendix : the B3 section is empty, the legend in figure 12 has a typo (100,000 to 150,000 is probably 1,000,000 to 1,050,000)

**Significance**
This is an important paper, which addresses a longstanding issue with neural networks. The parity and maze examples are very compelling. There seem to be many options for generalizing this approach, by varying the encoders and decoders, and the archtecture of the internal blocks. A discussion of these future options would improve the paper significance.

**Overall**
A very good paper on an important subject. It could be greatly improved by running a few experiments on inference, trying small changes in the architecture, and detailing a little morethe very conclusive experiments on parity integration and maze paths in the main paper. A clear accept.

**Time Spent Reviewing:**

6

---

> ### Author Response · Authors · 2021-08-10
> **Thank you for the input, our response to each point is as follows.**
>
> (1, 2) Multiple reviewers pointed out that they find the terms “easy” and “hard” to be ambiguous.  We have revised the draft to use the terms “larger” and “smaller” when those terms are applicable so as to minimize ambiguities and to be as clear as possible.
>
> Regarding chess:  You are correct that Chess presents complexities that are not present in mazes and prefix sums, and that our reported gains are more modest.  While the performance boost we observe by unrolling on chess puzzles is less than one point, we find it quite interesting that recurrent nets outperform feedforward nets on this problem (by a wide margin), and also that unrolling increased the number of problems the recurrent network could solve (even if by a modest amount) rather than resulting in model failure because of the stark difference between the training and testing regime.
>
> (3) When we increase the number of recurrences, the raw output at later iterations does eventually diverge from the solution, however we observe that the confidence decreases as well.  For this reason, the maximum confidence exit rule results in strong performance even when very large numbers of iterations are used.  We will include plots with very large numbers of iterations in our updated draft.
>
> (4) For in-distribution test accuracy, on prefix sums, mazes, and chess achieve >99.9%, >97%, and 91% on unseen small problems, respectively (using the same number of iterations as used during training). When we apply more iterations, we see only slight drops for in-distribution test accuracy (tenths of a percent), but no improvement. (5) Likewise, we have trained models on larger instances and tested them on smaller cases, and we find that the models can solve smaller examples in fewer iterations than were used at training. For example, when models are trained to compute prefix sums on 44-bit data in 10 iterations, they can get 100% of the 16-bit test examples correct after only 4 iterations. We have added these results and a discussion of them to the paper.
>
> (6) The architectures were chosen for simplicity’s sake. Normalization in recurrent modules is tricky as feature statistics on different iterations vary, and so sharing batch norm parameters (running mean and variance) between iterations does not result in good performance. We find that data normalization in the form of making the inputs zero-mean does help training slightly for prefix sums, and this technique is employed in our experiments. We have updated the paper to make this more clear.
> We’ve also added to the discussion of future work and further address the architectural choices.

---

> > ### Comment · Reviewer_XL39 · 2021-08-25
> > **Thank you for the clarifications**
> >
> > Thank you for the explanations. Your observations on the number of iterations are very interesting.

---

### Official Review · Reviewer_Yvok · 2021-07-16

**Rating:** 6
**Confidence:** 4

**Summary:**

The paper shows that recurrent neural networks, unlike feedforward ones, can generalize to harder tasks than they were trained on by increasing the number of recurrences during inference for tasks that allow such sequential reasoning.

**Limitations And Societal Impact:**

Yes.

**Main Review:**

The main numerical results from the paper are interesting, by showing that while RNNs underperform feedforward nets of the same effective depth, running an RNN for more iterations at test time can lead to vastly improved performance on harder tasks.

The visualization results of iteratively getting closer to the solution in combination of the raw performances are indeed suggestive that more recurrences leads to more reasoning.

It would have been interesting however to compare the performances of these recurrent architectures to significantly deeper feedforward networks as well as networks trained on the harder version of the tasks as control.

While the plots seems to suggest increasing recurrences usually helps, the Table 3 results seem to be showing saturation - the recurrent network drops 7% accuracy in performance while the feedforward gains nearly 3%. This suggests both that it would be worth checking deeper feedforward networks, as well as the fact that RNNs might be running into some sort of ceiling that should be explored here.

**Time Spent Reviewing:**

3.5

---

> ### Author Response · Authors · 2021-08-10
> **Thank you for the feedback.**
>
> We agree that the comparisons you mentioned are interesting, and we have thus added the following results to the latest draft of the paper.
>
> Significantly deeper feed forward models cannot generalize from easy to hard as well as recurrent ones. For example, we extended the experiments in Figure 4 to include feed-forward models with depths 132, 264, and 528. These models all fit the training data (32-bits) well, but the best performance on 40-bit data is 53% and on 44-bit data is 27% -- neither is attained by the deepest models lending confidence that the performance would not increase with even more depth. As far as training on the harder versions of the tasks, i.e. measuring in-distribution accuracy, we have found there is little difference between recurrent and feed forward models. As examples, both classes of models for prefix sums and mazes achieve essentially 100% on harder data not seen during training (>99% and >97% respectively).
>
> To follow up on your comments about Table 3, we note that in general, the performance continues to rise for recurrent models, but it does not for feed-forword. For example, on new trials that have been added to Table 3, we find that recurrent models with effective depths of 32 and 36 achieve 40.5% and 50.5% on large mazes, respectively. The feedforward models of the same effective depths only get 25.8% and 26.5% of the larger mazes correct.

---

> > ### Comment · Reviewer_Yvok · 2021-08-25
> > **Thank you for the answers**
> >
> > I would like to thank the authors for the responses, particularly on the deeper models and the results in Table 3.

---

### Decision · Program_Chairs · 2021-09-27

**Decision:**

Accept (Poster)

**Comment:**

This paper presents a genuinely interesting result, namely that by increasing the effective depth of a network through the repetition of learned dimensionality-preserving transformations within the network, the resulting test-time network obtains better extrapolatory performance without further training. It shows empirically on a diverse-enough set of toy-ish tasks to pique interest. I can see this paper generating quite a bit of discussion in a certain part of the NeurIPS community, in particular with regard to its application to Transformer-style architectures for which such an idea of adaptive depth could be easily applied.

There is a huge variance in scores, but thankfully, there was quite a bit of discussion, and the reviews are all fairly detailed (especially those in support of the paper). I'll be honest, I simply don't understand the argument made by Reviewer iKnp in favour of rejection. While it's true that the question "Can You Learn an Algorithm?" has been addressed by methods such as Neural GPUs, Turing Machines, Stacks etc circa 2014/2015, the matter is certainly not closed or solved in any substantial way, and a contribution such as that made by this paper is welcome. Aside from this point, I don't see a convincing argument in favour of rejection in Reviewer iKnp's comments.

Reviewer idGE offers a more substantial critical reading of the paper, which the authors should pay more heed to. I agree with some if this reviewer's point regarding the framing of the contribution, and being careful with the notion of hardness of problems. I urge the authors to consider fairly substantial edits to be a bit more diligent and conservative with their claims here, but at the end of the day would not consider rejecting the paper on this basis of this critique alone.

Reviewer Yvok seems satisfied with the author responses to their review, but has not updated their score past 6. However, I don't really see what outstanding issues they have that would prevent them from continuing in their support of the paper.

Finally, Reviewer XL39 has written substantially to champion the paper's acceptance, echoing however the issues raised by Reviewer idGE regarding terms such as "hard" which are used a little casually. I again recommend the authors take the task of editing the paper seriously to incorporate this feedback.

While on the face of it, this paper is borderline when it comes to scores, when weighing the reviewer recommendation by the strength of the arguments, and the detail of the reviews and discussions, I find that this paper is unambiguously acceptable and a strong contribution. It may not be perfect, but it will have good potential to influence others' work, and will add to the diversity of topics at the conference.